# AI-assisted discovery of potent FGFR1 inhibitors *via* virtual screening and *in silico* analysis

Ram Lal (Swagat) Shrestha[1,2,3ᴼ], Ashika Tamang[1,2ᴼ], Sandeep Poudel Chhetri[2,4ᴼ], Nirmal Parajuli[1,2], Manila Poudel[2], Shiva M. C.[1,2], Aakar Shrestha[2], Timila Shrestha[1,2], Samjhana Bharati[1,2], Binita Maharjan[1,2], Bishnu P. Marasini[2,3,5*‡], Jhashanath Adhikari Subin[ID][2,6*‡]

1 Department of Chemistry, Amrit Campus, Tribhuvan University, Lainchaur, Kathmandu, Nepal, 2 Kathmandu Valley College, Syuchatar Bridge, Kalanki, Kathmandu, Nepal, 3 Institute of Natural Resources Innovation, Kalimati, Kathmandu, Nepal, 4 Central Department of Physics, Tribhuvan University, Kirtipur, Kathmandu, Nepal, 5 Nepal Health Research Council, Ministry of Health and Population, Ramshah Path, Kathmandu, Nepal, 6 Bioinformatics and Cheminformatics Division, Scientific Research and Training Nepal P. Ltd., Bhaktapur, Kathmandu, Nepal

ᴼ These authors contributed equally to this work.
‡ BPM and JAS also contributed equally to this work.
* subinadhikari2018@gmail.com (JAS); bishnu.marasini@gmail.com (BPM)

## Abstract

Fibroblast growth factor receptor 1 (FGFR1) is recognized as an oncogene that fosters tumor development, playing a vital role in cancer progression. This has established it as a promising target for cancer drug development. However, existing FGFR1 inhibitors are often limited by drug resistance and lack of specificity, emphasizing the need for more selective and potent alternatives. To address this challenge, the present study employed an AI-driven virtual screening approach, integrating molecular docking (MD) and molecular dynamics simulations (MDS) to discover novel FGFR1 inhibitors. A voting classifier integrating three machine learning classifiers was utilized to screen 10 million compounds from the eMolecules database, leading to 44 promising candidates with a prediction probability exceeding 80%. MD identified compound with PubChem Compound Identifier (CID) 165426608 (−10.8 kcal/mol) as the highest-scoring ligand, while compounds with CID 145940129 (−9.8 kcal/mol), CID 131910163 (−9.4 kcal/mol), CID 155915988 (−9.2 kcal/mol), and CID 132423733 (−9.1 kcal/mol), exhibited binding affinities comparable to or slightly lower than that of the native ligand (−10.4 kcal/mol). MDS further revealed that all these compounds, except CID 131910163, maintained structural stability with time. Thermodynamic stability assessment confirmed the spontaneity and feasibility of their complex formation reactions with negative $\Delta G_{BFE}$ values ranging from −21.87 to −12.76 kcal/mol. Decomposition of binding free energy change further provided key stabilizing residues. The heatmaps and histograms of the interaction over the full 200 ns simulation period highlighted the prominent interaction profiles. Structural similarity analysis of the four MDS-stable compounds displayed the dice similarity

**Data availability statement:** All relevant data are within the paper and its Supporting Information files. Further raw data can be made available on request.

**Funding:** The author(s) received no specific funding for this work.

**Competing interests:** The authors have declared that no competing interests exist.

scores of 0.200000 to 0.452830 with known FGFR1 inhibitors. Additionally, the $pIC_{50}$ prediction using a voting regressor indicated promising $pIC_{50}$ values (7.07 to 7.47), highlighting their potential as hit candidates for further structural optimization and therapeutic development. Further, this study underscores the efficiency of machine learning-based virtual screening and *in silico* analysis as a cost-effective and reliable strategy for accelerating hit drug discovery from large datasets, even with limited resources and time.

## 1. Introduction

Cancer represents a critical global health challenge, accounting for one in every six deaths worldwide. In 2022, around 20 million individuals were newly diagnosed with cancer, and approximately 9.7 million deaths were attributed to cancer-related diseases [1]. Despite significant advancements in different therapeutic approaches such as chemotherapy, hormone therapy, and immunotherapy, cancer mortality rates remain alarmingly high. This is mainly because of the complex genetic and phenotypic diversity of cancer, and the emergence of drug-resistant phenotypes [2].

The fibroblast growth factor receptor (FGFR) signaling axis plays a crucial role in transducing signals that govern various cellular processes, including proliferation, angiogenesis, differentiation, embryonic development, migration, organogenesis, and survival [3]. Members of the fibroblast growth factor receptor family, including FGFR1, frequently undergo genomic alterations such as mutations, amplifications, and gene fusions across various cancer types [4]. FGFR1, in particular, has been extensively studied and recognized as an oncogene that fosters tumor development, underscoring its critical role in cancer progression [5]. Overexpression of FGFR1 has been observed in cancers such as breast [6,7], lung [8], ovarian [9,10], bladder [11], prostate [12,13], and gastric cancers [14], among others. Consequently, targeting FGFR1 for cancer therapy has become an appealing therapeutic strategy [15]. To date, drugs such as Regorafenib, Nintedanib, Sorafenib, Lenvatinib, Erdafitinib, Pemigatinib, Infigratinib, and Futibatinib have received FDA approval for FGFR1 inhibition [16–18]. These inhibitors work by reducing FGFR1 activity, which is often overexpressed in certain cancers, thereby impeding tumor growth. However, the efficacy of these inhibitors is limited due to challenges like drug resistance and lack of specificity [19]. Therefore, the development of novel inhibitors with enhanced effectiveness and reduced side effects remains a significant challenge.

Traditional drug discovery methods rely heavily on *in vivo* experiments and *in vitro* screening, which are both costly and labor-intensive [20]. Preclinical drug discovery constitutes approximately one-third of the drug development expenses and usually takes nearly five and a half years [21,22]. The high failure rate during drug development further exacerbates the costs. As a result, methodologies that can reliably predict success at early stages are critically valuable. Computer-aided drug design (CADD) has emerged as a transformative approach in this domain [23]. By employing *in silico* techniques, CADD accelerates drug discovery and reduces the time

required for identifying leads and introducing new drugs. These methods also enable the prediction of biological activity for chemical compounds against specific targets [24]. In this study, an Artificial Intelligence (AI)-driven virtual screening approach of millions of molecules was adopted to advance FGFR1-targeted drug discovery [25,26]. By leveraging AI's ability to analyze vast datasets and predict drug efficacy in a relatively short time span, this study aims to streamline the drug development process and increase the likelihood of identifying alternate treatments [27]. Fig 1 outlines the detailed workflow adopted in this study.

## 2. Materials and methods

### 2.1. Data collection and curation

The Chemical European Molecular Biology Laboratory (ChEMBL) database was used to obtain the Simplified Molecular Input Line Entry System (SMILES) representations and half-maximal inhibitory concentration ($IC_{50}$) values for 2,153 FGFR1 inhibitors [28]. The $IC_{50}$ value represents the concentration of a compound required to inhibit a specific biological process or activity by 50% which serves as a preliminary guide for selecting efficient and biologically active molecules. After filtering entries without $IC_{50}$ values, retaining bioactivity data measured in nanomolar (nM), and removing duplicates, 1876 data points remained. The $IC_{50}$ values were transformed into $pIC_{50}$ values using negative logarithms to standardize the data. Lipinski's Rule of Five (RO5) was applied to assess drug-likeness and exclude less potent compounds [29,30], resulting in 1523 data points for model training. Radar plots depicting the physicochemical properties of the filtered dataset are shown in Fig 2.

### 2.2. Model building and database screening

Molecular fingerprints [31], encoded as numerical vectors or bit-strings, facilitate rapid similarity evaluations critical for virtual screening [31,32], structure-activity relation studies, and chemical space mapping [33]. Using the RDKit toolkit [34], fingerprints from SMILES entries were computed, and the dataset was classified into 813 active and 710 inactive compounds (1523 total) using a $pIC_{50}$ threshold of 7.0, as a cut-off ranging from 5 to 7 has been recommended [26]. Based on the Morgan3 protocol, which employs 2048 bits as a circular fingerprint [35], machine learning models were constructed

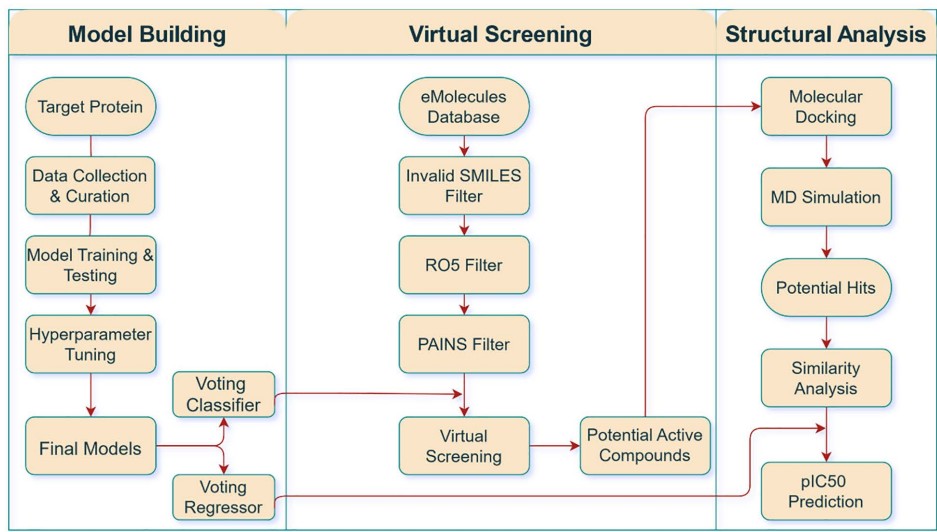

**Fig 1. Detailed workflow of the study.**

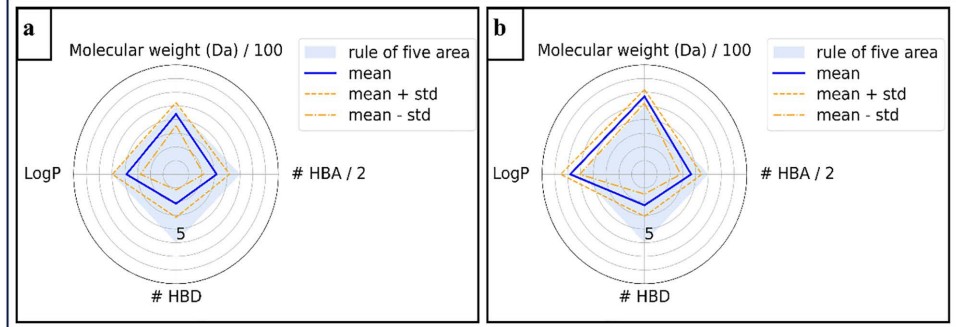

**Fig 2. Physicochemical radar plots of the compound in the dataset that (a) fulfill RO5 and (b) violate RO5.**

using Scikit-learn and XGBoost. The Morgan3 fingerprints (radius = 3) encode molecular features extending up to three bonds from each atom, allowing the capture of broader substructural patterns that may play a vital role in determining biological activity. Scikit-learn is a versatile machine-learning library that provides a diverse range of algorithms for classification, regression, clustering, and dimensionality reduction tasks [36]. XGBoost is an advanced library tailored for the fast and scalable execution of gradient-boosting algorithms [37]. Twenty classification models were trained, and the best-performing models were fine-tuned to create a voting classifier, amplifying accuracy and robustness in comparison with the individual models [38]. A similar approach was applied for building a voting regressor. The voting classifier was then employed to screen 10 million compounds from the eMolecules database [39,40]. Compounds with invalid SMILES, those violating the Rule of Five (RO5), and Pan-Assay Interference Compounds (PAINS) were excluded prior to screening.

Classification models' performance was evaluated using accuracy, precision, sensitivity, specificity, and Area Under Curve (AUC) metrics, calculated based on the confusion (error) matrix [41]. Regressors were assessed according to mean absolute error (MAE), root-mean-squared error (RMSE), and R2 scores [42]. Multiple steps were used systematically for model training and database screening to identify potential molecules.

### 2.3. Molecular docking calculations

The 44 potential inhibitors with prediction probabilities above 80% obtained from the screening of 10 million compounds from the eMolecules database were selected as hit ligands. The 3D structures of 35 compounds available in the PubChem database (https://pubchem.ncbi.nlm.nih.gov/) [43] were retrieved in SDF format, while the remaining 9 compounds were drawn using their SMILES strings. The molecular formulas were verified using the Avogadro program (v1.2.0) [44] after adding the hydrogen atoms. Energy minimization was carried out using the UFF force field with 5000 steps employing a conjugate gradient algorithm, ensuring energy convergence at $1.0 \times 10^{-8}$ kcal/mol. This process was repeated until the global minima was reached. The bond orders, including double bond positions, were examined, and steric hindrance or stress was removed. Finally, the optimized ligands were converted to PDBQT format with Gasteiger charges using AutoDock Tools [45].

The 3D crystal structure of FGFR1 (PDB ID: 4ZSA, DOI: https://doi.org/10.2210/pdb4ZSA/pdb) with X-ray crystallographic resolution of 2.00 Å was obtained from the RCSB database (https://www.rcsb.org/) [46]. Missing amino acid residues were repaired using the SwissModeling server (https://swissmodel.expasy.org/) [47], where model_01 of template 5B7V.1.A (global model quality estimate: 0.88, qualitative model energy analysis with distance constraints: 0.83 ± 0.05) was selected due to its 100% sequence identity. The finalized protein structure was converted to PDBQT format with the addition of polar hydrogens and Kollman charges using the AutoDock Tools. The apo form of the protein was then utilized as the target for computational analyses.

The molecular docking calculations of the ligands with the FGFR1 protein were done using the user-friendly software AutoDock Vina [48], with the same protocol as outlined by Phunyal et al. [49], with slight modifications of parameters. A grid box size of (50, 50, 50) Å³, the grid center at (x: 5.060, y: − 0.501, z: 16.013), an energy range of 4, and 20 number of modes were used with an exhaustiveness (converged) of 64. The five protein-ligand complexes with the top binding affinities were saved in pdb format and utilized for MDS. The binding interaction between the protein and ligand was visualized using the PyMOL [50] program and the protein-ligand interaction profiler (PLIP) [51] web server.

## 2.4. Molecular dynamics simulations

The GROMACS (version 2021.2) software [52] was used to simulate the protein-ligand complex, with the CHARMM36 force field [53] applied to the receptor, while the ligand parameters were derived from the SwissParam server [54]. The system was solvated using the TIP3P water model in a triclinic box with a 12 Å spacing at the sides, to prevent any unwanted effects caused by repetition of the simulation box. Neutralization was achieved by adding counter ions, followed by the inclusion of an isotonic NaCl solution (0.15 M). Equilibration was carried out in four steps with each of 1 ns at 310 K and 1 bar, with the first two using the NVT ensemble and the last two using NPT. The V-rescale thermostat, which is a modified version of the Berendsen method, was used for the temperature coupling, while pressure coupling was applied through the isotropic Parrinello-Rahman approach. The final 200 ns production run, with a 2 fs step size, was performed without constraints on the protein-ligand complex. Additional parameter details for different system setups can be found in the literature [55–58]. The complex was centered and analyzed using GROMACS built-in modules to obtain geometric parameters such as snapshots, root mean square deviation (RMSD) of the ligand and protein backbone, root mean square fluctuation (RMSF), radial pair distribution function (RPDF), radius of gyration ($R_g$), and solvent-accessible surface area (SASA).

The thermodynamic stability of the protein-ligand complexes was evaluated using the Molecular Mechanics Poisson-Boltzmann Surface Area (MMPBSA) binding free energy calculations [59], following the parameters as utilized by Shrestha et al. [60]. A 20 ns equilibrated segment from the 200 ns molecular dynamics trajectory was used for this purpose. Calculations were carried out using the MMPBSA module [61], which applies the Poisson–Boltzmann solvation model. Binding free energy was computed at 100 ps intervals to assess the stability, spontaneity, and feasibility of complex formation over time. The spontaneity and viability of the forward reaction were evaluated based on the sign of the free energy changes. To investigate the contribution of individual amino acid residues to binding free energy change, decomposition analysis was performed on the same 20 ns equilibrated segment using the g_mmpbsa tool [62], which allowed the decomposition of the total binding energy to identify key stabilizing and destabilizing residues. The associated gmx_MMPBSA_ana subprogram was used for data analysis and visualization.

Additionally, to understand residue-level interaction dynamics over the entire course of the simulation, amino acid interaction heatmaps and histograms were generated using the full 200 ns trajectory (20,000 frames).

## 2.5. Similarity analysis

Based on the principle that structurally similar compounds often share chemical and biological properties, a similarity analysis between potential and known FGFR1 inhibitors [63] was conducted. The relevance of this analysis is closely tied to the nature of structure-activity relationships (SARs) that define biologically active molecules, serving as key factors for the success of ligand-based virtual screening, regardless of the methods employed [64]. Using Morgan2 fingerprints, the similarity maps based on the dice similarity metric were generated, highlighting structural features influencing biological activity [65,66]. Morgan2 fingerprints (radius = 2) encode molecular features extending up to two bonds, emphasizing localized structural variations. In similarity maps, Morgan2 is beneficial for highlighting key local substructures that impact activity, enhancing the interpretability of the visualization.

## 2.6. Computational resources

All the calculations, including plot generation, were executed on high-performance multiprocessor systems. The machine learning computations were conducted on a system featuring 96 cores, 256 GB of RAM, a 16 GB GPU accelerator, and running Ubuntu 20.04 LTS. Meanwhile, MD and MDS were performed on a system with 24 cores, a 24 GB GPU accelerator, and running Ubuntu 20.04 LTS. The analysis and the visualization of the data were done on a personal computer with Windows 11 operating system.

## 3. Results and discussion

### 3.1. Model evaluation and screening results

Twenty classification models were evaluated, and their performance metrics are summarized in Table 1.

The Support Vector Classifier (SVC), ExtraTreesClassifier (ET), and Extreme Gradient Boosting Classifier (XGB) demonstrated superior accuracy and AUC scores, leading to their integration into a voting classifier with a soft voting mechanism. The soft voting mechanism is an ensemble learning technique that predicts the final class by averaging the probability estimates from multiple models and selecting the class with the highest mean probability [67]. For SVC, the parameters were set to C = '2.0' and probability = 'True'; for ET, n_estimators = '400', criterion = 'log_loss', and max_features = 'log2'; for XGB, n_estimators = '1000', max_depth = '5', and learning_rate = '0.04'; all other parameters were set to their default values.

Fig 3 presents confusion matrices for individual and voting classifiers, while Table 2 summarizes the five-fold cross-validation results.

ROC curves in Fig 4 illustrate the excellent discrimination ability of these models.

**Table 1. General performance of 20 different classification models.**

| Models | Accuracy | AUC |
|---|---|---|
| SVC | 0.88 | 0.88 |
| ExtraTreesClassifier | 0.88 | 0.88 |
| NuSVC | 0.88 | 0.87 |
| RandomForestClassifier | 0.88 | 0.87 |
| XGBClassifier | 0.88 | 0.87 |
| LGBMClassifier | 0.85 | 0.85 |
| CalibratedClassifierCV | 0.85 | 0.85 |
| BaggingClassifier | 0.85 | 0.85 |
| LogisticRegeression | 0.85 | 0.85 |
| Perceptron | 0.84 | 0.84 |
| PassiveAggressiveClassifier | 0.84 | 0.84 |
| KNeighborsClassifier | 0.83 | 0.83 |
| DecisionTreeClassifier | 0.82 | 0.82 |
| SGDClassifier | 0.82 | 0.82 |
| ExtraTreeClassifier | 0.82 | 0.82 |
| LinearSVC | 0.81 | 0.81 |
| AdaBoostClassifier | 0.81 | 0.81 |
| BernoulliNB | 0.81 | 0.81 |
| NearestCentroid | 0.80 | 0.80 |
| RidgeClassifierCV | 0.80 | 0.80 |

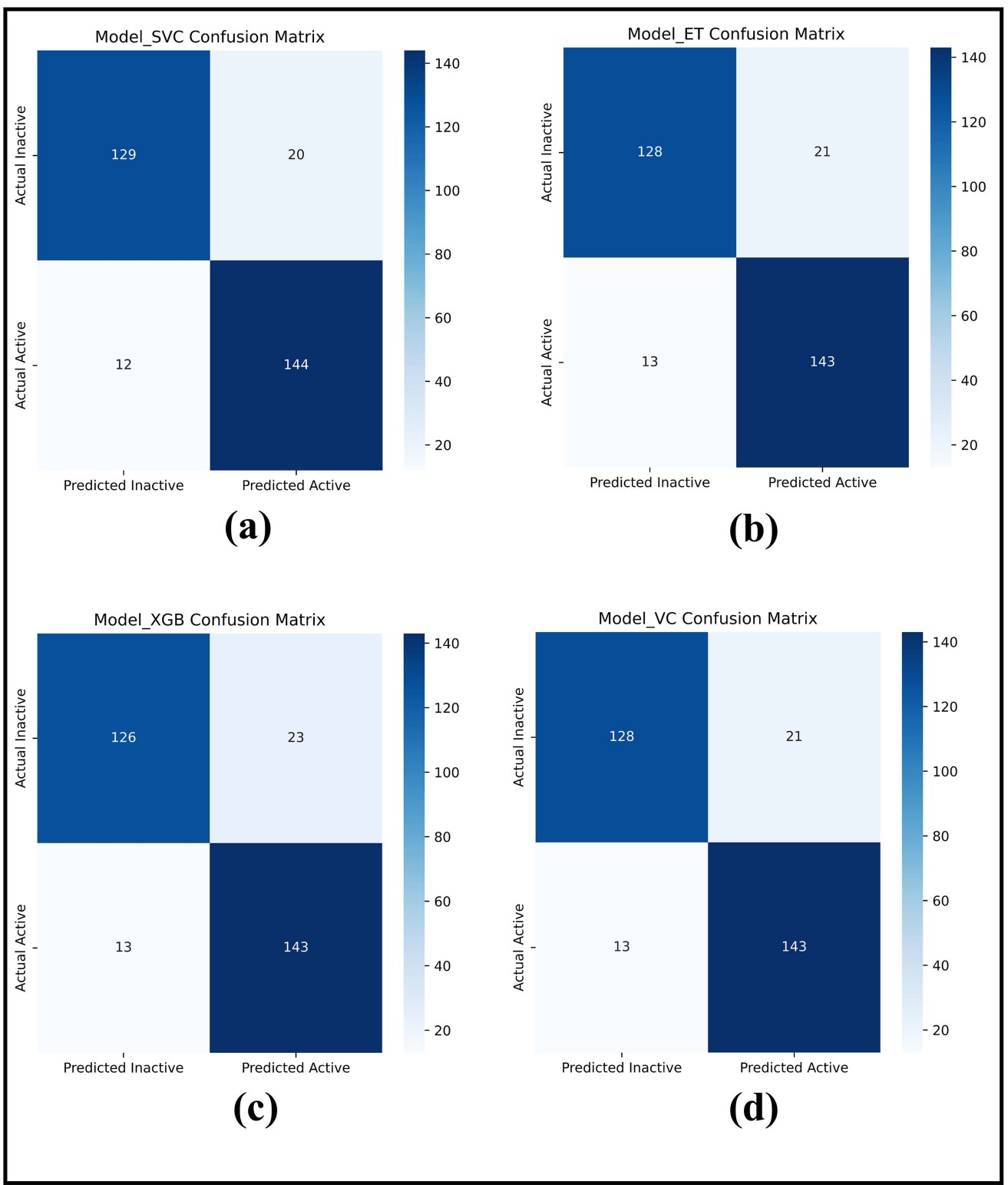

**Fig 3. Confusion matrix for (a) SVC; (b) ExtraTreesClassifier; (c) XGB Classifier; (d) Voting Classifier.**

**Table 2. Five-fold cross-validation of individual classifiers and voting classifier using five parameters.**

| Model | Accuracy | Precision | Sensitivity | Specificity | AUC |
|---|---|---|---|---|---|
| SVC | 0.88±0.01 | 0.88±0.01 | 0.90±0.02 | 0.86±0.02 | 0.95±0.01 |
| ET | 0.88±0.01 | 0.88±0.02 | 0.90±0.03 | 0.86±0.03 | 0.95±0.01 |
| XGB | 0.88±0.01 | 0.88±0.02 | 0.90±0.01 | 0.86±0.03 | 0.94±0.01 |
| VC | 0.89±0.01 | 0.89±0.02 | 0.91±0.01 | 0.87±0.02 | 0.95±0.01 |

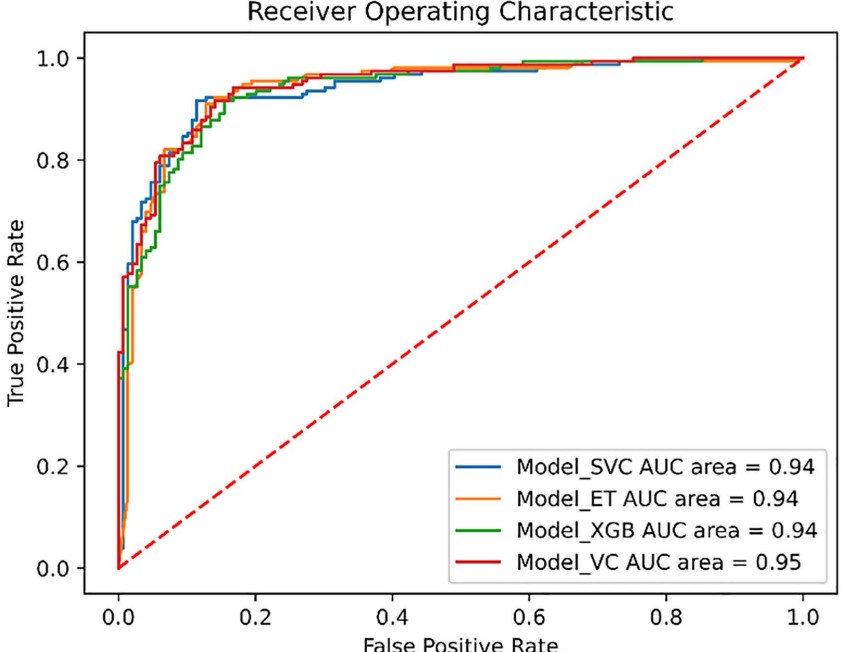

**Fig 4. ROC curves of the individual models and the voting classifier.**

The AUC scores show that all the classifiers have excellent discrimination abilities between active and inactive compounds. Additionally, a voting classifier was tested on an external test set, which consisted of FDA-approved selective inhibitors of FGFR1- Erdafitinib, Pemigatinib, Infigratinib, and Futibatinib. The model classified all these drugs as active with high prediction probabilities (> 90%), further demonstrating the reliability of our model. Based on these results, the voting classifier was used to screen the eMolecules database, identifying 44 compounds with prediction probabilities above 80% as potential active inhibitors of FGFR1 protein.

### 3.2. Docking score comparison and interaction analysis

The molecular docking protocol was validated by docking the native ligand into the apo protein's active site. The pose of the docked native ligand from the molecular docking was superimposed on its pose from the crystal structure, as shown in Fig 5, resulting in a heavy atom RMSD of 0.397 Å (< 2 Å) [68]. This confirmed the parameters, algorithm, and ligand poses, justifying the numerical method adapted with the capability of reproducing the natural process.

The binding affinities and poses of ligands interacting with the FGFR1 protein were obtained from molecular docking calculations. The 44 ligands obtained by the virtual screening of 10 million compounds were subjected to molecular docking calculations against the FGFR1 receptor to assess their potential for competitive inhibition. Ligand M34 exhibited the

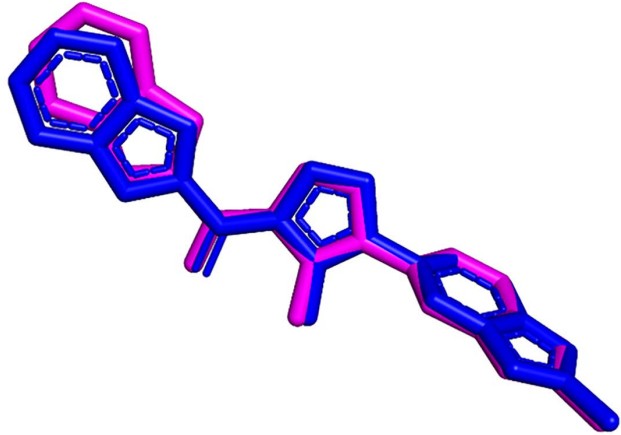

**Fig 5. Superimposed image of native ligand (blue) from the crystal structure with the docked native ligand (magenta) from molecular docking calculations (a heavy atom RMSD = 0.397 Å).**

highest binding affinity (−10.8 kcal/mol), outperforming the native ligand (−10.4 kcal/mol). Additionally, four ligands, M29 (−9.8 kcal/mol), M26 (−9.4 kcal/mol), M32 (−9.2 kcal/mol), and M28 (−9.1 kcal/mol), demonstrated binding affinities comparable to or slightly lower than that of the native ligand. The binding affinity of 44 ligands is presented in Table 3 along with their molecular formula, PubChem chemical identifier (CID), and parent ID.

The mode of interactions was studied for the complexes of M26, M28, M29, M32, and M34 with the protein. The 3D representations of the complexes, as shown in Fig 6, demonstrated that all top five ligands were bound at the catalytic site of the protein, suggesting the competitive inhibitors.

To better understand the details at the molecular level, the bonding interactions between the top five docked ligands and key amino acid residues, along with the distances, were studied (Table 4) and the interaction profiles are presented in Fig 7. The interaction analysis revealed several key hydrophobic interactions between the ligands and the protein's amino acid residues, along with hydrogen bonds.

The molecular interaction analysis revealed that all ligand-protein complexes (M26-M34) were primarily stabilized by hydrophobic interactions, especially with residues Leu27, Val35, Val104, Leu173, and Phe185, which appeared frequently across the top complexes and the native complex, except M29-complex. Among these, residues Leu27, Leu173, and Phe185 were the most consistently conserved hydrophobic residues, underlining their importance in ligand anchoring. Notably, residue Asp184 (3–4 Å) contributed to hydrogen bonding in M26 and M32, while M26 further exhibited additional hydrogen bonds with Phe185 (3.93 Å) and Gly186 (4.08 Å), suggesting a stabilizing role of Asp184 that was not observed in the native complex.

In contrast, the native complex formed distinct hydrogen bonds with Glu105 (2.80 Å) and Ala107 (2.87, 2.98 Å), interactions not replicated by the ligand-bound complexes. Despite this, the native complex shared hydrophobic contacts with residues Leu27, Val104, Leu173, and Phe185, which were also involved in complexes of M26, M28, M32, and M34, indicating partial overlap in binding site occupation. M29-complex exhibited a unique interaction pattern, including hydrogen bonds (Lys57: 3.06 Å, Asp195: 2.90 Å), a π-stacking interaction with Phe32, and distinct hydrophobic residues, implying a different binding orientation. Additionally, M28 uniquely formed a halogen bond with residue Ile88 (3.51 Å) *via* a fluorine atom.

Overall, hydrophobic interactions emerged as the key driving force for ligand binding, with residues Leu27, Val104, Leu173, and Phe185 serving as conserved contributors across multiple complexes. In addition, hydrogen bond distances

**Table 3. Binding affinities for 44 ligands with the FGFR1 protein, along with that of the native ligand. (Bold-faced was chosen for MDS).**

| Compounds | | PubChem CID | Parent ID | Molecular Formula | Binding Affinities (kcal/mol) |
|---|---|---|---|---|---|
| **Ligands** | **M34** | **165426608** | **338098121** | **$C_{36}H_{41}N_5O_6$** | **−10.8** |
| | **M29** | **145940129** | **323961687** | **$C_{38}H_{47}FN_6O_9$** | **−9.8** |
| | **M26** | **131910163** | **290025896** | **$C_{21}H_{25}N_5O_2$** | **−9.4** |
| | **M32** | **155915988** | **331142671** | **$C_{19}H_{22}N_4O_3$** | **−9.2** |
| | **M28** | **132423733** | **316221773** | **$C_{24}H_{28}FN_5O$** | **−9.1** |
| | M33 | 163358414 | 337753544 | $C_{24}H_{25}N_5O_2$ | −9.0 |
| | M5 | 15944647 | 22060461 | $C_{21}H_{18}Cl_2N_4O_2$ | −8.9 |
| | M17 | 75410789 | 48548835 | $C_{22}H_{22}FN_3O_5$ | −8.9 |
| | M12 | 75410713 | 48548681 | $C_{21}H_{22}FN_3O_3$ | −8.8 |
| | M19 | 75410793 | 48548843 | $C_{22}H_{21}F_4N_3O_3$ | −8.8 |
| | M30 | 145936598 | 323969661 | $C_{41}H_{48}FN_7O_8$ | −8.8 |
| | M35 | NR | 338535583 | $C_{26}H_{35}FN_4O_2$ | −8.8 |
| | M36 | NR | 342779808 | $C_{24}H_{25}F_3N_4O_4$ | −8.8 |
| | M4 | 45160386 | 20467665 | $C_{26}H_{32}N_4O_5$ | −8.7 |
| | M13 | 75410731 | 48548717 | $C_{22}H_{25}N_4O_2$ | −8.6 |
| | M16 | 75410782 | 48548821 | $C_{22}H_{22}FN_3O_5$ | −8.6 |
| | M20 | 75410795 | 48548847 | $C_{24}H_{25}FN_4O_3$ | −8.6 |
| | M11 | 75410711 | 48548677 | $C_{21}H_{24}FN_3O_2$ | −8.5 |
| | M14 | 75410740 | 48548735 | $C_{20}H_{23}FN_4O_2$ | −8.5 |
| | M8 | 75410670 | 48548595 | $C_{20}H_{20}F_4N_4O_2$ | −8.4 |
| | M7 | 75410654 | 48548563 | $C_{19}H_{20}F_2N_4O_2$ | −8.3 |
| | M9 | 75410675 | 48548605 | $C_{21}H_{22}FN_3O_4$ | −8.2 |
| | M24 | 75410803 | 48548863 | $C_{21}H_{23}FN_4O_4$ | −8.1 |
| | M6 | 42453839 | 25992257 | $C_{26}H_{37}N_5O_3S$ | −8.0 |
| | M15 | 75410742 | 48548739 | $C_{18}H_{22}FN_5O$ | −8.0 |
| | M2 | 45177117 | 20428857 | $C_{24}H_{35}N_5O4$ | −7.9 |
| | M22 | 75410800 | 48548857 | $C_{22}H_{24}FN_3O_5S$ | −7.9 |
| | M1 | 45168627 | 20392252 | $C_{29}H_{36}N_4O_4$ | −7.8 |
| | M18 | 75410790 | 48548837 | $C_{20}H_{21}FN_4O_3$ | −7.8 |
| | M21 | 75410797 | 48548851 | $C_{19}H_{22}FN_5O_3$ | −7.7 |
| | M44 | 168463113 | 345870006 | $C_{20}H_{31}N_7O_3$ | −7.7 |
| | M25 | 75410804 | 48548865 | $C_{19}H_{22}FN_5O_3$ | −7.6 |
| | M3 | 45177479 | 20430493 | $C_{26}H_{34}N_4O_3$ | −7.5 |
| | M31 | 145942272 | 324002419 | $C_{20}H_{23}N_7O_3$ | −7.5 |
| | M10 | 75410691 | 48548637 | $C_{18}H_{22}FN_5O_2$ | −7.4 |
| | M27 | 155880121 | 313775290 | $C_{20}H_{28}N_4O_3$ | −7.4 |
| | M43 | NR | 345224621 | $C_{23}H_{33}N_5O_4S$ | −7.1 |
| | M39 | NR | 344364342 | $C_{24}H_{35}N_5O_3$ | −7.0 |
| | M23 | 75410801 | 48548859 | $C_{18}H_{19}FN_4O_3S$ | −6.9 |
| | M38 | NR | 344358704 | $C_{26}H_{39}N_5O_3$ | −6.8 |
| | M40 | NR | 344365110 | $C_{26}H_{39}N_5O_3$ | −6.6 |
| | M42 | NR | 344373816 | $C_{25}H_{36}ClN_5O_3$ | −6.5 |
| | M37 | NR | 344352563 | $C_{24}H_{35}N_5O_3$ | −6.1 |
| | M41 | NR | 344373304 | $C_{23}H_{32}ClN_5O_3$ | −6.0 |
| **Native ligand** | LWJ | 66555680 | – | $C_{20}H_{16}N_6O$ | −10.4 |

Note: NR = Not Recorded

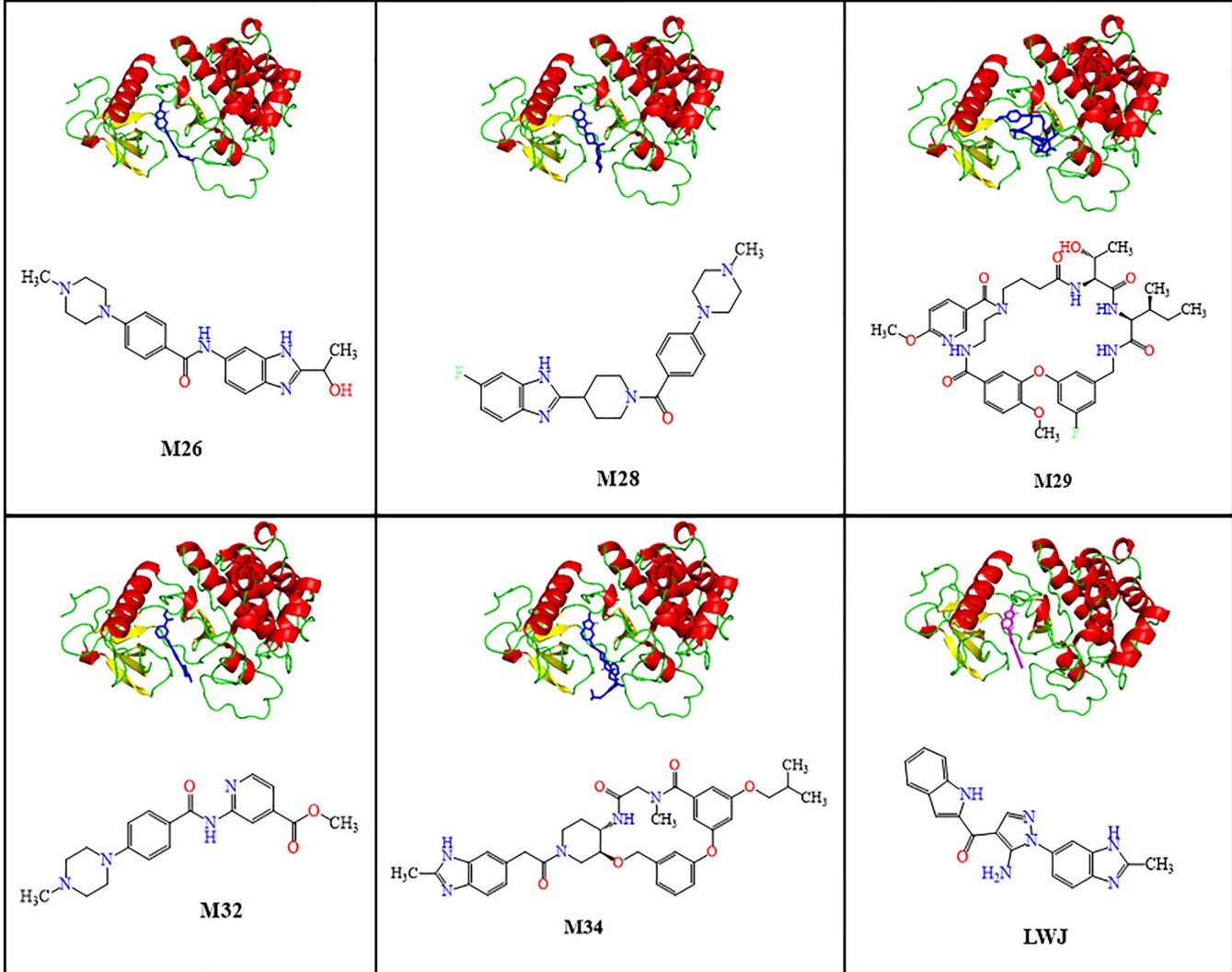

**Fig 6. 3D representations (left) of docked ligands at the catalytic pocket of the protein in complexes and the molecular structures (right) of the ligands.**

ranging from 2.90 to 4.03 Å indicated strong to moderate binding affinity, as shorter bond lengths generally correlate with stronger interaction [69]. While most protein-ligand complexes exhibited similar hydrophobic interaction patterns with the native complex, M29 displayed a distinct binding profile, which may influence its receptor modulation potential.

### 3.3. Adduct stability with time (spatial and energetic)

Understanding the spatial and energetic stability of the adduct is crucial for evaluating the inhibitory potential of ligands on the FGFR1 protein. To achieve this, MDS was performed for 200 ns. The structural and interaction stability of the protein-ligand complexes for the top five ligands was analyzed by examining various time-dependent parameters. The spontaneity and feasibility of the complex formation reactions for the top five ligands were determined in terms of changes in binding free energy. Both geometric and thermodynamic parameters are discussed in the following sections.

**Table 4. Types of interactions between the top five docked ligands and key amino acid residues in the protein-ligand complexes.**

| Complexes | Types of interactions | Amino acid residues (Distance, Å) |
|---|---|---|
| **Protein-M26 complex** | Hydrogen bond | Asp184 (3.51, 3.53, 3.53), Phe185 (3.93), Gly186 (4.08) |
| | Hydrophobic | Leu27 (3.62, 3.67, 3.78), Val35 (3.30, 3.57, 3.93), Tyr106 (3.75, 3.88), Leu173 (3.72), Phe185 (3.31) |
| **Protein-M28 complex** | Halogen bond (Fluorine) | Ile88 (3.51) |
| | Hydrophobic | Val35 (3.58, 3.79), Ile88 (3.58), Val104 (3.67, 3.83), Leu173 (3.71), Ala183 (3.80) |
| **Protein-M29 complex** | Hydrogen-bond | Lys57 (3.06), Asp195 (2.90) |
| | Pi-stacking | Phe32 (4.58) |
| | Hydrophobic | Ala31 (3.69), Ile194 (3.88), Lys198 (3.76), Thr200 (3.73), Pro206 (3.32, 3.87) |
| **Protein-M32 complex** | Hydrogen-bond | Asp184 (3.00, 4.03) |
| | Hydrophobic | Leu27 (3.52, 3.71), Val35 (3.79), Val104 (3.83), Leu173 (3.79) |
| **Protein-M34 complex** | Hydrogen bond | Leu27 (3.75) |
| | Hydrophobic | Pro26 (3.55), Leu27 (3.76), Val35 (3.47, 3.59), Val104 (3.58), Leu173 (3.97), Phe185 (3.60), Thr201 (3.76) |
| **Protein-Native complex** | Hydrogen-bond | Glu105 (2.80), Ala107 (2.87, 2.95) |
| | Hydrophobic | Leu27 (3.53, 3.92), Lys57 (3.81), Ile88 (3.78), Val104 (3.65), Leu173 (3.71), Phe185 (3.66) |

**3.3.1. Structural stability assessment.** The binding of the ligand to the protein can induce structural alterations in both the protein and the ligand, which may affect the stability of the complex and hence the inhibition mechanism [70]. The stability of the top five ligand complexes was evaluated by analyzing various computational metrics from MDS trajectories, which provided insight into the structural integrity of the protein-ligand complexes. The metrics include the study of ligand pose (snapshot), RMS deviation of ligands and protein backbone relative to protein backbone, RMS fluctuation of the α-carbon atoms, RPDF, $R_g$, and SASA which are discussed next.

***Dynamic insights into ligand behavior at the protein active site through MDS:*** Snapshots were taken at various time intervals during the MDS to examine the orientation and position of the docked ligands, providing insight into the stability of the complex's geometry over time. Detailed images of the top four complexes at five distinct instances are presented in Fig 8.

Snapshots taken at 1, 50, 100, 150, and 200 ns showed that most ligands remained at the same location but with variations in orientation, except for a few cases. For the M28-complex, the ligand exhibited distinct rotational motion starting at 1 ns, accompanied by a slight upward position shift from 100 ns till the end due to translational movement. The protein backbone displayed some motion, particularly in the α-helix and loops (on the right side) from 50 ns onward, with subtle movement in the central β-sheet. In the case of M29-complex, the ligand underwent significant positional and orientational changes at the active site, with pronounced rotation at 50 ns and minimal rotational motion thereafter. From 100 ns onward, the ligand moved slightly upward, eventually returning to its position at 50 ns by 200 ns. The protein backbone showed the movement of the right α-helix (absent before but observed after 50 ns till the end), along with motions in the left β-sheet and central loops. In the M32-complex, the ligand depicted minimal delocalization and rotation until 100 ns, followed by rotation and a slight downward shift in the position along with the noticeable motion of the α-helix and loops (150 ns). In the M34-complex, the ligand remained stable with some rotation for the first 50 ns. After this period, it slightly shifted upward along with the protein backbone, maintaining minimal rotational and translational movement until the end of the simulation. The protein backbone showed notable dynamics, including the disappearance of the left α-helix after 1 ns and upward movement of the top lying α-helix (100 ns). The loops fluctuated throughout the simulation. The M26-complex was excluded from Fig 8 as the ligand showed displacement from the orthosteric site after 150 ns, suggesting weak binding or instability that compromised complex integrity. This indicated that the binding affinity of

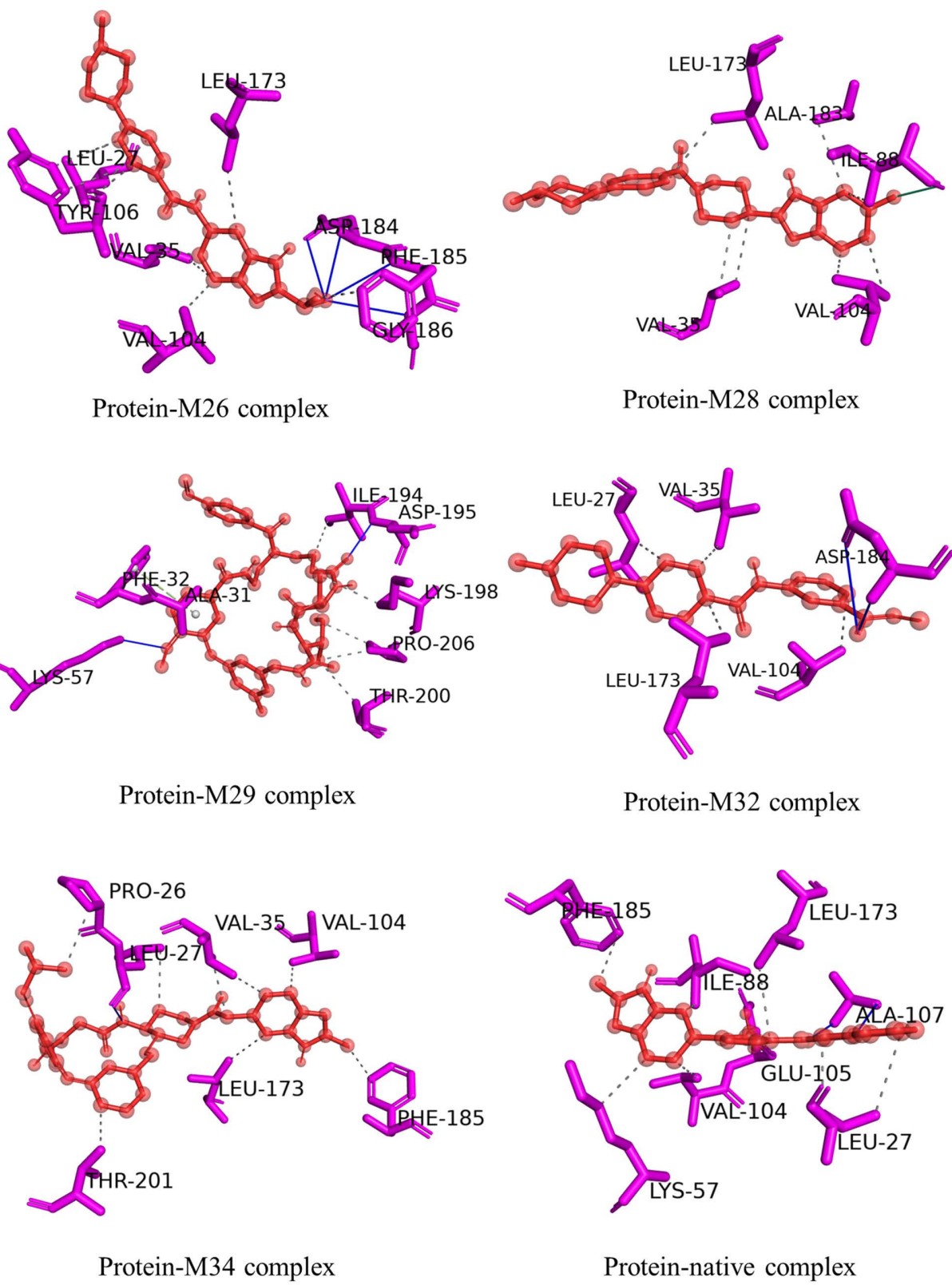

**Fig 7. Molecular-level interaction profiles of top docked complexes along with that of the native complex.**

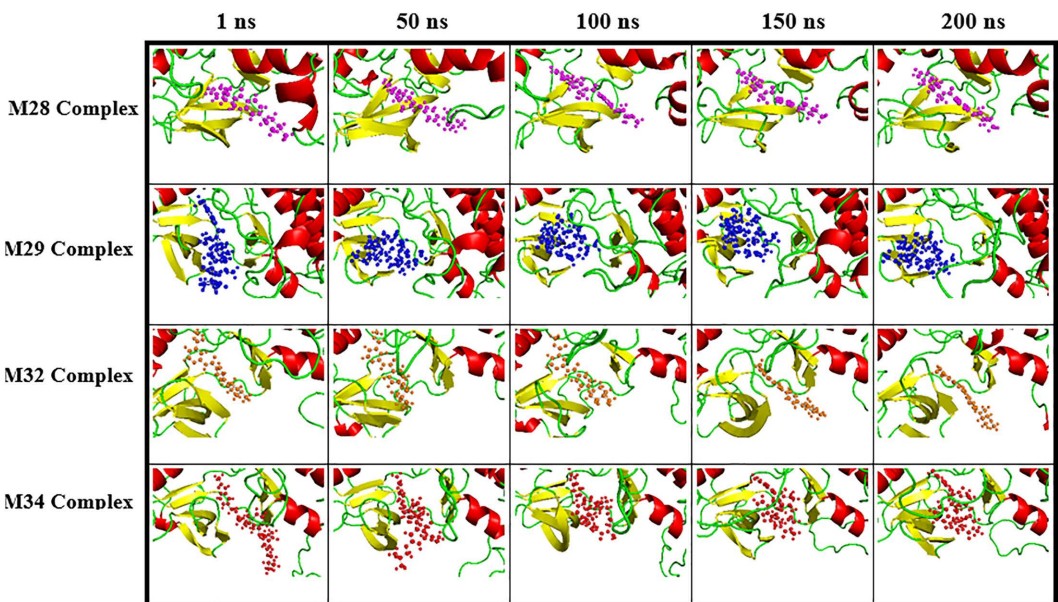

**Fig 8. Ligand's positions and orientations within the protein's active site across four different complexes at five distinct instances, as extracted from the MDS trajectories.**

−9.4 kcal/mol, even though better than that observed for M32 and M28 complexes, was not sufficient to retain the pose and position at the active site. This implies that MD does not necessarily provide information about the stability of the complexes.

Periodic monitoring of adducts' dynamical behavior provided valuable insights into molecular evolution, which could be linked to specific structural descriptors. Overall, the results demonstrated that the ligand's pose and the protein backbone's structural integrity were nearly preserved across the top four complexes (M28, M29, M32, and M34) with minimal structural changes and no major disruptions, suggesting the stability of the complexes. On the other hand, the M26-complex was unstable due to ligand displacement. These findings can be correlated with the RMSD and RPDF curves, which will be discussed next.

**RMSD of ligand and protein backbone in the complex:** The stability and dynamic behavior of the protein-ligand complexes were analyzed by examining the RMSD of the ligands and protein backbones. The RMSD for both the ligands and protein backbones with respect to the protein backbone was calculated from the MDS trajectories of various complexes and is displayed in Figs 9 and 10, respectively.

The RMSD of ligands relative to the protein backbone (Fig 9) provides insight into the extent of the conservation of the pose over time. The RMSD profiles of M28 (blue) and M29 (red) in their respective complexes exhibited smooth trajectories, with average RMSD of 0.38±0.11 nm and 0.56±0.08 nm, respectively. A slight increase in fluctuation was observed for M29 after 155 ns, attributed to an orientation shift of the ligand beyond 150 ns, as illustrated in (Fig 8). In the case of M34 (magenta), the RMSD trajectory was relatively flat after 95 ns, whereas M32 (maroon) depicted a moderate RMSD curve throughout the simulation period with some fluctuation after 120 ns. The fluctuation before 95 ns in the M34 complex and after 120 ns in M32 can be corroborated with the ligand's orientation as seen previously in snapshots (Fig 8). The average RMSD for M34 and M32 was determined to be 0.63±0.16 nm and 0.49±0.16 nm, respectively. Conversely, ligand M26 (green) initially displayed a stable trajectory up to 170 ns, followed by a sharp increase in RMSD, reaching approximately 2.0 nm. This trend suggested an unstable nature of the complex, which aligns with the interpretation from the snapshots.

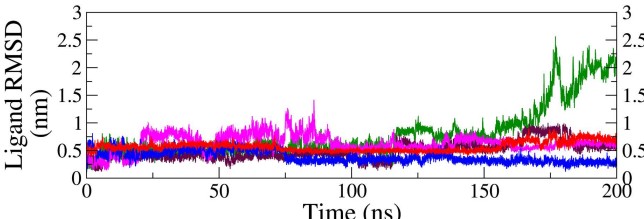

**Fig 9. RMSD curves of top ligands relative to protein backbone; M26-complex (green), M28-complex (blue), M29-complex (red), M32-complex (maroon), and M34-complex (magenta).**

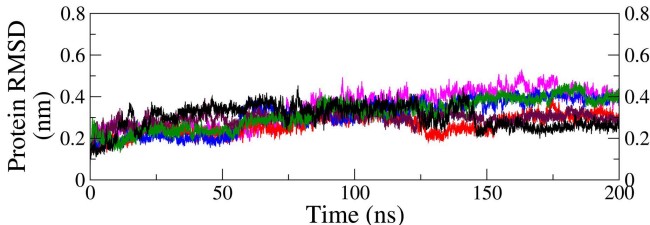

**Fig 10. RMSD curves of protein backbone with respect to protein backbone in holo form along with the protein backbone in apo form (black); M26-complex (green), M28-complex (blue), M29-complex (red), M32-complex (maroon), and M34-complex (magenta).**

The backbones (Fig 10) of M28 (blue) and M34 (magenta) followed similar trajectories, with average RMSD of 0.31 ± 0.07 nm and 0.34 ± 0.08 nm, respectively, closely matching that of the apo protein (black = 0.30 ± 0.05 nm). In contrast, the backbones of M29 (red) and M32 (maroon) exhibited slightly lower average RMSD of 0.26 ± 0.04 nm and 0.29 ± 0.03 nm, respectively. The observed spikes in the RMSD plots corresponded to minor backbone adjustments, as depicted in the snapshots (Fig 8). These findings suggest that the protein backbone remained largely stable across the four complexes, indicating that ligand binding had minimal impact on its overall structure compared to the apo form. Since the M26 was unstable, the protein backbone of the M26-complex was not discussed.

From the analysis of the RMSD profiles, it was found that all ligands except M26 were bound at the protein's active site till the end, narrowing the selection of top candidates from five to four. Among the four, ligands M28 and M29 resulted in the most stable complexes, as reflected by their low and consistent RMSD. M34-complex exhibited good stability, whereas M32-complex demonstrated moderate stability. In contrast, the complex with M26 showed significant instability, with a sharp increase in RMSD after 170 ns, indicating a loss of stable binding, and therefore, protein backbone analysis was omitted. The protein maintained the sturdy geometry across the top four complexes, capable of holding the ligand at its catalytic site. Hence, four ligands, except M26, could potentially inhibit the functioning of the FGFR1 protein.

**Radial pair distribution function (RPDF):** The radial pair distribution function (RPDF) describes how the distance between two entities varies over time. In its reduced form [g(r)], it represents the probability of finding the ligand's center of mass at a distance r from the protein's center of mass [71]. Fig 11 represents the RPDF plots for different protein-ligand complexes, which have been derived from the MDS trajectories.

The RPDF plot revealed distinct binding behavior of the ligands relative to the protein's center of mass. M32 (maroon), M28 (blue), and M34 (magenta) displayed two peaks at different distances, whereas M29 (red) exhibited a single peak. For M32, two peaks with a taller one at *ca.* 0.9 nm and a shorter one at *ca.* 1.2 nm were observed. Similarly, a tall peak at *ca.* 1.1 nm and a short peak at *ca.* 1.4 nm were observed for M28. The two different peaks indicated occupancy at two distinct locations, with the longer peak suggesting a preference for a shorter ligand-protein distance. On the other hand,

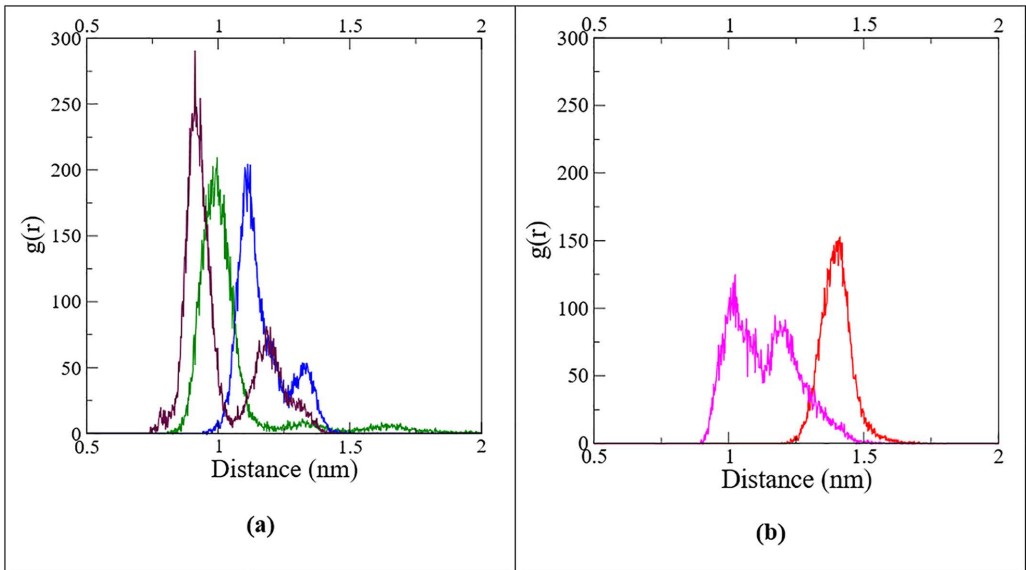

**Fig 11. Radial pair distribution function between the center of mass of ligands ((a) M26 = green, M28 = blue, M32 = maroon; (b) M29 = red, M34 = magenta) and protein's center of mass in various complexes retrieved from the MDS trajectories; a single sharp peak indicates the ligand's localization and a double peak signifies occupation at two distinct positions during the MDS period.**

M34 displayed two peaks of comparable height at *ca.* 1.0 nm and 1.2 nm, implying that it occupied two distinct locations within the protein's active site for most of the simulation period. The presence of these peaks supported the minor variations in ligand position and orientation within the complex, as observed in Fig 8. In contrast, the M29 (red) complex exhibited a single peak at *ca.* 1.4 nm, indicating the localization of the ligand's center of mass relative to the protein's center of mass throughout the simulation. The occurrence of RPDF maxima at *ca.* 1.0 nm for the top four ligands indicates that the orthosteric site remained occupied throughout most of the simulation period. These results indicate that after binding to the orthosteric pocket of the receptor protein, the ligands remained largely localized within the site, possibly inhibiting the protein's regular function. Thus, the RPDF analysis effectively evaluated ligand stability over time, reinforcing earlier conclusions drawn from structural snapshots.

For M26 (green), a single peak was observed at *ca.* 1.0 nm, but the presence of further smaller broad peaks afterward indicated the delocalization, which supported the instability of the ligand, as previously noted in snapshots and RMSD analysis. Since ligand M26 exhibited instability, further geometrical parameter analysis for this ligand was not conducted.

**Fluctuation of α-carbon in the protein backbone of the complex:** The root mean square fluctuation (RMSF) of the α-carbon atoms was calculated from the MDS trajectory to identify the flexible and rigid regions of the FGFR1 protein after ligand binding. The RMSF curves for the five ligand-protein complexes, along with that of the apo form (Fig 12), displayed a similar nature of the plot.

The RMSF was below *ca.* 0.8 nm for all top four complexes, whereas it was *ca.* 1.0 nm for the apo form, indicating the stability of protein geometry [72]. Higher RMSF were observed at the terminal ends and within three specific loop regions around residues 45, 130, and 200. The increased flexibility in these regions can be attributed to the absence of α-helix or β-sheet structures, which typically restrict molecular motion and reduce degrees of freedom [55]. Since these flexible regions do not play a significant role in ligand interactions or disruptions, their high RMSF does not indicate structural instability. Therefore, for clarity in the plot, only residues ranging from 12 to 300 were considered, excluding the terminal ends. The RMSF profiles indicated that the α-carbon atom fluctuations exert minimal influence on the ligand's binding

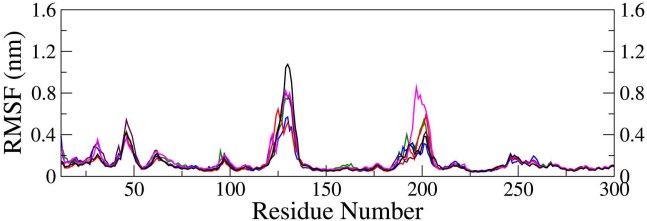

**Fig 12. Root mean square fluctuation of α-carbon atoms in protein backbone of five complexes (in holo form) along with that in the apo form (black); M28-complex (blue), M29-complex (red), M32-complex (maroon), and M34-complex (magenta).**

affinity within the active site. Consequently, the stability of the adduct remained unaffected, which potentially may lead to the inhibition of protein activity.

**Gyradius ($R_g$) and solvent-accessible surface area (SASA):** The gyradius ($R_g$), derived from the MDS trajectory, was used to evaluate the protein's compactness and backbone conformational changes. It represents the average distance of the macromolecule's components from its central axis, which is a crucial indicator of the stability of the protein-ligand complex [73]. The $R_g$ plot for the protein backbone of the four ligand-protein complexes (Fig 13) revealed similar stability patterns across all systems, with $R_g$ values ranging from *ca.* 2.00 to 2.15 nm, indicating no significant structural expansion or contraction during the simulation period. In contrast, the M29 (red = 2.05 ± 0.01 nm) and M32 (maroon = 2.06 ± 0.02 nm) complexes displayed some fluctuations, particularly between 60 ns and 145 ns, similar to that of the apo form (black = 2.06 ± 0.02 nm) and M34 (magenta = 2.07 ± 0.02 nm). A pronounced fluctuation in the case of the M34 complex can be correlated with minor changes in ligand orientation and α-helix positioning at 100 ns (Fig 8). Overall, the $R_g$ of the top four complexes closely matched the apo form, suggesting no significant receptor expansion or shrinkage upon ligand binding. These findings indicate that the protein maintained structural integrity even after ligand binding, suggesting that the ligands may contribute to target protein inhibition.

Solvent-accessible surface area (SASA) is a crucial parameter for evaluating protein-solvent interactions, as it measures the exposure of protein residues to water molecules [74]. Changes in SASA can influence the protein's structure, dynamics, and function [75]. SASA analysis was conducted to evaluate the effect of ligand binding on the conformational behavior of the FGFR1 protein over 200 ns MD simulations (Fig 14). The SASA ranged from 155 to 185 nm$^2$ for most complexes, showing similar trends to that of the apo form (black), with an exception for the M34 complex (magenta = 172.94 ± 4.48 nm$^2$) which exhibited slightly higher SASA due to minor surface adjustments after 80 ns. The M28 (blue = 166.00 ± 3.32 nm$^2$), M29 (red = 166.83 ± 2.59 nm$^2$), and M32 (maroon = 168.59 ± 3.75 nm$^2$) complexes displayed comparable average SASA to the apo form (165.47 ± 3.13 nm$^2$), supporting the stable surface geometry. The minimal variations observed (below 5 nm$^2$) suggest that ligand binding did not significantly alter the protein's hydrophobic regions

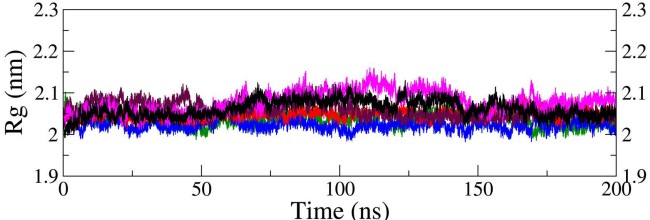

**Fig 13. Radius of gyration curves for protein backbone in top five ligand complexes along with the apo form (black): M28-complex (blue), M29-complex (red), M32-complex (maroon), and M34-complex (magenta); extracted from MDS trajectories.**

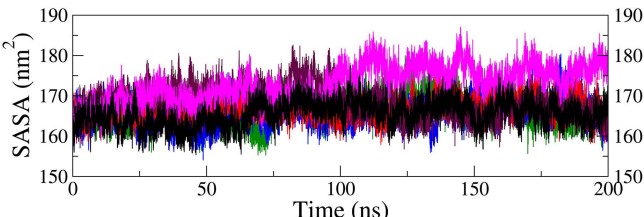

**Fig 14. Solvent-accessible surface area plots for protein backbone in top five ligand complexes along with the apo form (black): M28-complex (blue), M29-complex (red), M32-complex (maroon), and M34-complex (magenta); extracted from MDS trajectories.**

or shape, ensuring consistent solvent accessibility and reinforcing the structural stability of the complexes for most of the cases, as discussed previously.

**3.3.2. Thermodynamic stability assessment of the protein-ligand complexes.** The spontaneity and feasibility of complex formation reactions were evaluated by analyzing the binding free energy changes in the equilibrated segment of the MDS trajectory (20 ns, 200 frames) for the top four (MDS stable) ligand adducts, as outlined in Table 5. The table reflects the degree of spontaneity in the complex formation reactions from the discrete protein and ligand. The negative $\Delta G_{BFE}$ ($\Delta G_{BFE} < 0$) signifies the spontaneity of the complex formation reaction, and a smaller value corresponds to higher stability [76].

All top-ranked protein-ligand complexes exhibited negative $\Delta G_{BFE}$ (ranging from −21.87 to −12.76 kcal/mol), affirming the spontaneity and feasibility of the complex formation reactions. Among them, the M34-complex demonstrated the highest thermodynamic stability, with the lowest $\Delta G_{BFE}$ of −21.87 ± 3.98 kcal/mol. Analysis of the thermodynamic components revealed that the solvent contribution from the Poisson-Boltzmann model (pb) posed a significant destabilizing influence across all the complexes. Nevertheless, this adverse effect was effectively mitigated by substantial positive contributions from electrostatic (el), van der Waals (vdW), and non-polar (np) interactions. These findings suggest that the top four ligands exhibit a natural propensity to associate with the FGFR1 receptor, forming energetically favorable and stable complexes throughout the simulation period. Notably, M34 stood out as the most stable adduct based on its overall energy profile.

To further dissect the energetic contributions at the residue level, decomposition analysis was performed. M29 exhibited the most favorable binding energy (−13.57 ± 2.00 kcal/mol), primarily stabilized by hydrophobic residues such as Phe32, Gly33, and Val35. Minor destabilizing effects were noted from residues Asp184 and Lys57. Similarly, M28 showed

**Table 5. Binding free energy changes (kcal/mol) and their components for protein-ligand complexes extracted from the equilibrated segment of the 20 ns MDS trajectories.**

| Thermodynamic Components | Binding Free Energy Change (kcal/mol) | | | |
|---|---|---|---|---|
| | M28-complex | M29-complex | M32-complex | M34-complex |
| $\Delta E_{vdw}$ | −44.43 ± 2.48 | −41.17 ± 3.67 | −29.49 ± 2.62 | −51.58 ± 3.56 |
| $\Delta E_{el}$ | −18.89 ± 5.27 | −23.75 ± 8.74 | −9.07 ± 4.11 | −21.72 ± 5.82 |
| $\Delta E_{pb}$ | 50.05 ± 5.34 | 52.86 ± 11.42 | 29.28 ± 4.70 | 56.89 ± 5.27 |
| $\Delta E_{npolar}$ | −5.00 ± 0.16 | −4.75 ± 0.35 | −3.47 ± 0.19 | −5.46 ± 0.22 |
| **$\Delta G_{BFE}$** | **−18.28 ± 4.30** | **−16.81 ± 7.20** | **−12.76 ± 2.83** | **−21.87 ± 3.98** |

Note: data are expressed as average±SD

where, $\Delta G_{BFE}$ = Change in Gibb's binding free energy, $\Delta E_{npolar}$ = Non-polar energy change, $\Delta E_{pb}$ = Change in Poisson-Boltzmann solvation energy, $\Delta E_{el}$ = Electrostatic energy change, $\Delta E_{vdw}$ = van der Waals energy change

strong binding affinity (−10.12±1.98 kcal/mol), with stabilizing contributions from Asn111, Val35, and Leu173, while residues Lys57, Glu74, and Glu105 depicted unfavorable effects. The M34-complex displayed moderate binding energy (−9.87±2.00 kcal/mol), supported by residues Val35, Val104, Asn171, Leu173, and Ala183, and opposed by interactions with Lys57 and Asp184. In contrast, M32 demonstrated the weakest binding (−6.57±1.27 kcal/mol), with stabilizing hydrophobic contacts at residues Leu27, Val35, and Leu173, but significant destabilization from Asp184 and Lys57. Detailed residue-wise free energy contributions are presented in Supplementary Information S1 Table in S1 File. Corresponding bar plots and heatmaps highlighting the interactions between active site residues of FGFR and the respective ligands are shown in Supplementary Information S1-S5 Figs in S1 File, respectively. In these heatmaps, blue shades denote favorable (negative value) contributions, while red shades represent unfavorable (positive value) ones [62].

Overall, the results indicate that hydrophobic and polar residues, particularly Val35 and Leu173, played dominant roles in stabilizing the ligand-FGFR complexes. M29 emerged as the most promising ligand candidate based on per-residue energetic contributions, while M34 demonstrated the greatest thermodynamic stability based on $\Delta G_{BFE}$. Collectively, these findings suggest that all top ligands formed energetically stable complexes with FGFR1, with M34 and M29 showing particularly strong and favorable interactions. These results highlight their potential as promising FGFR1-targeted inhibitors, although further experimental validation is required to confirm their therapeutic efficacy.

### 3.4. Protein-ligand interaction heatmaps and histograms

To understand the residue-level binding dynamics, amino acid interaction heatmaps and histograms were generated for the 200 ns (20,000 frames) simulation period and provided in Supplementary Information S6-S13 Figs in S1 File. The analyses revealed that all four ligands complexed with the FGFR1 protein predominantly exhibited stable van der Waals (vdW) interactions, with sporadic hydrogen bonding observed in some cases.

In the M28-complex, the ligand maintained strong and consistent vdW interactions with residues Leu27, Glu105, Ala107, and Ser108. Among these, Ser108 and Glu105 accounted for the highest number of contacts, followed by Ala107and Leu27, suggesting their central role in ligand stabilization. M29 showed stable vdW contacts with residues Gly33, Gln34, and Met58, while transient interactions were observed with Gly28 and Glu29 residues. A hydrogen bond with residue Asn20 emerged after frame 13,600. Residues Gly33 and Gln34 were the most frequently engaged residues in the M29-complex, with moderate contacts involving residues Met58, Gly29, Gly28, and Asn202. In the M32-complex, the ligand demonstrated stable vdW contacts with residues Ala107 and Ser108 post frame 11300, along with moderate transient interactions involving residues Leu27 and Glu29. Residue Leu27 also formed intermittent hydrogen bonds, further contributing to ligand retention. The residue Ser108 was the most frequently interacting residue in this complex, followed by Ala107 and Leu27. For the M34-complex, stable vdW interactions were primarily observed with residues Glu29, Leu27, Gly18, and Arg170. Occasional hydrogen bonding was also noted with residue Leu27. The corresponding bar plot/histogram indicated residue Glu29 as the most engaged residue, followed by Leu27, with moderate contributions from Arg170, Glu105, and Gly28 residues.

Collectively, these findings underscore the importance of stable vdW contacts, particularly with residues Glu29, Gly33, Glu105, and Ser108 in securing ligand binding and maintaining complex stability. Their frequent engagement across multiple complexes highlights their potential relevance as key anchoring sites in FGFR1-ligand interactions.

### 3.5. Similarity measure analysis

The chemical similarity of four top candidate compounds (as suggested by MDS) against known FGFR1 drugs was evaluated. Fig 15 shows how structural modifications influence similarities, providing insights into the potential effectiveness of new compounds using a two-color scheme to highlight conserved and divergent features [77]. Green regions represent structural elements shared with reference drugs, suggesting retention of critical pharmacophores needed for FGFR1 inhibition, while pink areas indicate novel modifications or distinct scaffolds [66].

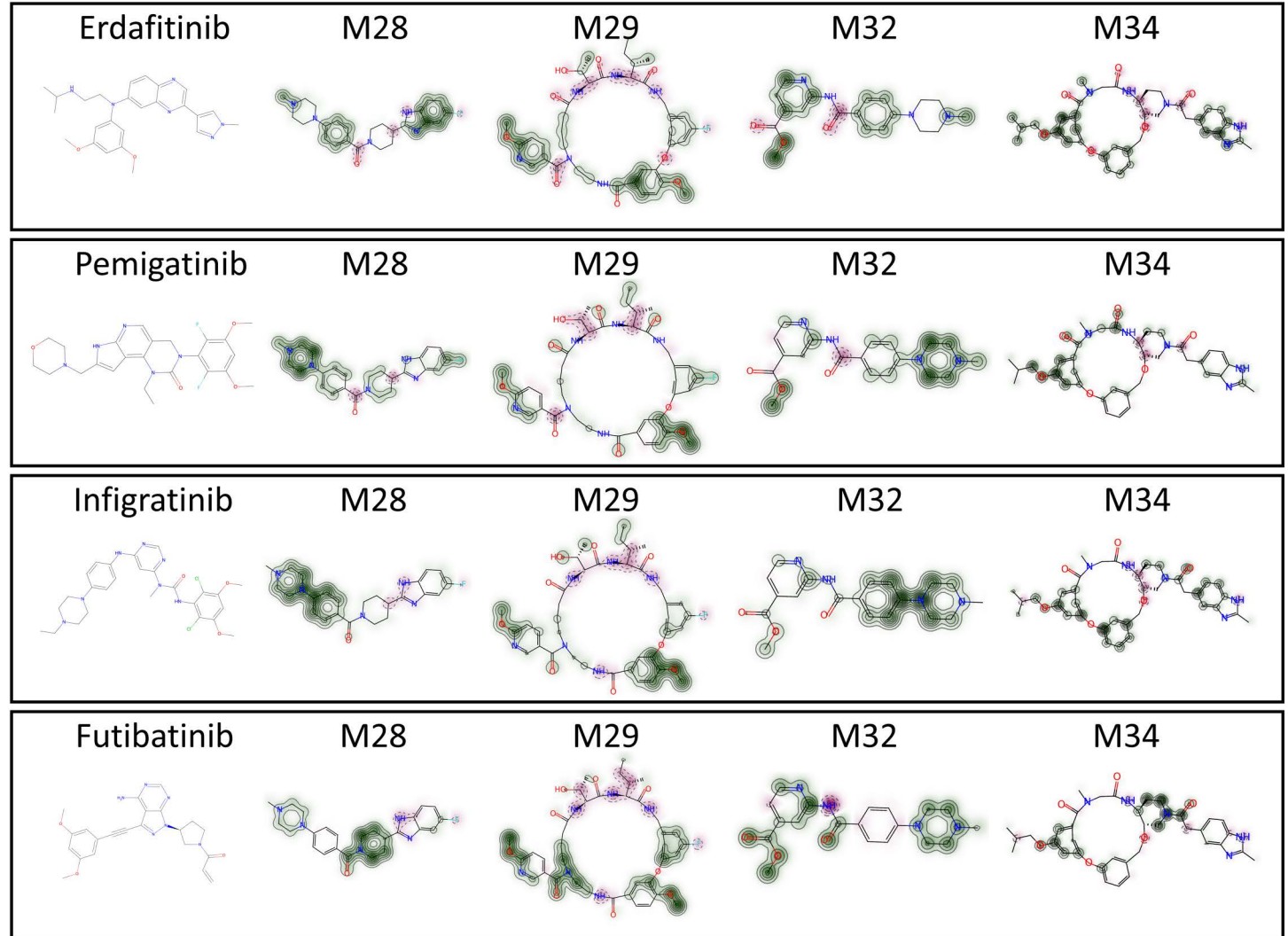

**Fig 15. Similarity analysis between known and potential inhibitors of FGFR1.**

The analysis revealed that M32 exhibits strong green overlap with infirgatinib, implying it likely maintains a similar binding mode and could serve as a high-priority candidate for lead optimization. Compounds M28 and M29 displayed a mix of green and pink regions, indicating partial structural conservation with opportunities for hybrid scaffold development to fine-tune selectivity or potency. In contrast, M34 showed prominent pink regions, suggesting unique structural features that may either introduce novel binding mechanisms or require further validation to confirm target specificity. The overall results suggest that M32 is the most promising candidate due to its strong structural similarity to known FGFR1 inhibitors, while M28-M34 offer opportunities for optimization or novel scaffold exploration.

The dice similarity index calculated between the known and potential inhibitors of FGFR1 is shown in Table 6.

The dice similarity index revealed that M32 has the strongest structural resemblance to infirgatinib (0.4532830), making it the most promising candidate. M28 showed moderate similarity across all inhibitors, with the highest value for infirgatinib (0.313043). M29 exhibited comparable similarity to erdafitinib (0.251656) and pemigatinib (0.294872), indicating shared structural features. Meanwhile, M34 had the lowest overall similarity, peaking with futibatinib (0.263158). Overall, these

**Table 6. Dice similarity index between the known and potential FGFR1 inhibitors.**

| Candidate Compounds | Erdafitinib | Pemigatinib | Infirgatinib | Futibatinib |
|---|---|---|---|---|
| M28 | 0.200000 | 0.260870 | 0.313043 | 0.263158 |
| M29 | 0.251656 | 0.294872 | 0.282051 | 0.232258 |
| M32 | 0.257426 | 0.283019 | 0.452830 | 0.266667 |
| M34 | 0.243243 | 0.235294 | 0.209150 | 0.263158 |

results suggest that all hit compounds are viable leads, with M32 standing out as the most promising, warranting further optimization to enhance potency.

## 3.6. Prediction of $pIC_{50}$ values

In addition to classification models, a regression model was used to estimate the $pIC_{50}$ values of the candidate compounds. Table 7 summarizes the performance metrics of the top 20 regression models.

Light Gradient Boosting Machine (LGBM), Hist Gradient Boosting (HGB), and Gamma Regressor (GR) were combined into a voting regressor for superior performance. For LGBM, the parameters were set to n_estimators = '600' and learning_rate = '0.05'; for HGB, max_iter = '400' and learning_rate ='0.05'; for GR, max_iter = '500' and alpha = '0.02'; all other parameters were set to their default values.

The five-fold cross-validation of individual regression models and the voting regressor is shown in Table 8.

Experimental versus predicted $pIC_{50}$ values are shown in Fig 16, while Table 9 compares these values for known and potential inhibitors.

**Table 7. General performance of 20 different regression models.**

| Model | R-Squared | RMSE |
|---|---|---|
| LGBMRegressor | 0.71 | 0.68 |
| HistGradientBoostingRegressor | 0.71 | 0.69 |
| GammaRegressor | 0.68 | 0.72 |
| TweedieRegressor | 0.68 | 0.72 |
| BayesianRidge | 0.67 | 0.73 |
| RandomForestRegressor | 0.67 | 0.73 |
| SVR | 0.66 | 0.75 |
| ElasticNetCV | 0.65 | 0.75 |
| PoissonRegressor | 0.65 | 0.75 |
| NuSVR | 0.65 | 0.75 |
| LassoLarsCV | 0.65 | 0.75 |
| LassoCV | 0.65 | 0.75 |
| BaggingRegressor | 0.64 | 0.76 |
| GradientBoostingRegressor | 0.63 | 0.78 |
| KNeighborsRegressor | 0.55 | 0.85 |
| PassiveAggressiveRegressor | 0.55 | 0.86 |
| HuberRegressor | 0.55 | 0.86 |
| RidgeCV | 0.52 | 0.88 |
| OrthogonalMatchingPursuitCV | 0.49 | 0.91 |
| ExtraTreesRegressor | 0.46 | 0.94 |

**Table 8. Five-fold cross-validation of individual regressors and voting regressor.**

| Model | R-Squared | MAE | RMSE |
|---|---|---|---|
| LGBM | 0.76±0.03 | 0.47±0.02 | 0.65±0.03 |
| HGB | 0.76±0.03 | 0.47±0.02 | 0.65±0.03 |
| GR | 0.74±0.04 | 0.50±0.02 | 0.66±0.04 |
| VR | 0.77±0.03 | 0.47±0.02 | 0.63±0.04 |

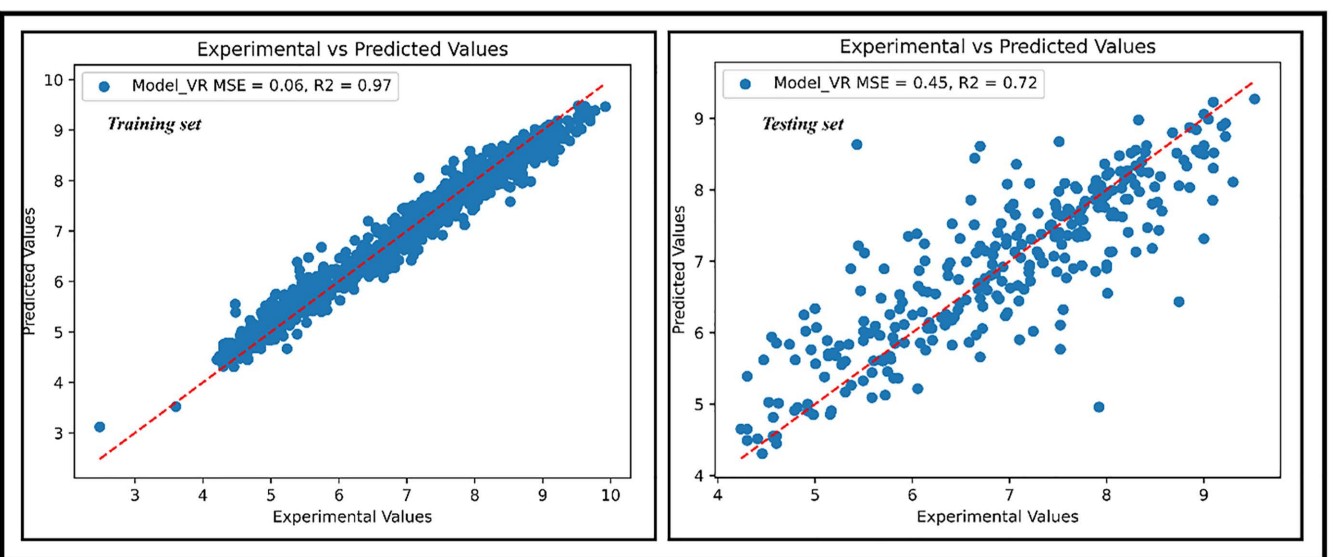

**Fig 16. Experimental versus predicted pIC$_{50}$ for training and testing set.**

**Table 9. Predicted and experimental pIC$_{50}$ values for known inhibitors and potential inhibitors of FGFR1.**

| Compound | Experimental pIC$_{50}$ | Predicted pIC$_{50}$ |
|---|---|---|
| Erdafitinib | 8.92 | 8.84 |
| Pemigatinib | 9.40 | 9.16 |
| Infigratinib | 9.04 | 8.16 |
| Futibatinib | 8.41 | 8.21 |
| M28 | – | 7.07 |
| M29 | – | 7.47 |
| M32 | – | 7.29 |
| M34 | – | 7.17 |

The predicted pIC$_{50}$ values of the hit candidate compounds (7.07–7.47), while lower than those of FDA-approved selective FGFR1 inhibitors, demonstrate promising lead-like potency. These values suggest that the compounds possess potential as FGFR1 inhibitors and could benefit from further structural optimization. Collectively, the findings support their candidacy for continued validation and development in FGFR1-targeted drug discovery.

Overall, the study identified four hit drug candidates as FGFR1 inhibitors by screening a large dataset. The combination of machine learning-based virtual screening with an *in silico* method has been found to accelerate the preliminary drug

discovery process, enabling efficient analysis of extensive datasets within a short period despite the resource constraints. This integration has not only improved accuracy but also ensured the reliable identification of high-potential drug candidates for FGFR1 inhibition, streamlining the selection for further studies.

## 4. Conclusion

The study demonstrates the effectiveness of combining AI-guided screening with molecular docking and dynamics simulations in identifying structurally stable and energetically favorable FGFR1 inhibitors. Four hit molecules were identified from a pool of 10 million molecules using this approach. The added insights from per-residue energy decomposition and long-term interaction histogram profiling offered valuable mechanistic insights into ligand-residue interactions, reinforcing the reliability of the identified candidates. These computational results offer a cost effective and quick foundation for further investigation. Future work will focus on pharmacophore modelling, structural optimization of hit compounds and their biological evaluation through *in vitro* and *in vivo* studies to support preclinical development.

## Supporting information

**S1 File. Supplementary data.** This file contains additional data supporting the findings of this study.
(DOCX)

## Acknowledgments

The author(s) hereby declare that Artificial Intelligence (AI) technology (ChatGPT) has been used during the preparation of the work to improve the readability and language of the manuscript. After using this tool/service, the author(s) reviewed and edited the content as needed and take(s) full responsibility for the content of the published article.

## Author contributions

**Conceptualization:** Jhashanath Adhikari Subin.

**Data curation:** Ram Lal (Swagat) Shrestha, Ashika Tamang, Sandeep Poudel Chhetri, Samjhana Bharati, Binita Maharjan.

**Formal analysis:** Bishnu P. Marasini, Jhashanath Adhikari Subin.

**Investigation:** Ram Lal (Swagat) Shrestha, Ashika Tamang, Sandeep Poudel Chhetri, Manila Poudel.

**Methodology:** Ram Lal (Swagat) Shrestha, Ashika Tamang, Sandeep Poudel Chhetri.

**Resources:** Ram Lal (Swagat) Shrestha, Timila Shrestha, Samjhana Bharati, Binita Maharjan.

**Software:** Ram Lal (Swagat) Shrestha, Binita Maharjan.

**Supervision:** Bishnu P. Marasini, Jhashanath Adhikari Subin.

**Validation:** Nirmal Parajuli, Manila Poudel.

**Visualization:** Nirmal Parajuli, Shiva M.C., Aakar Shrestha.

**Writing – original draft:** Ram Lal (Swagat) Shrestha, Ashika Tamang, Sandeep Poudel Chhetri.

**Writing – review & editing:** Bishnu P. Marasini, Jhashanath Adhikari Subin.

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
