## [Decision Letter · Decision Letter 0]

10 Jun 2025

PONE-D-25-21397AI-Assisted Discovery of Potent FGFR1 Inhibitors via Virtual Screening and In Silico AnalysisPLOS ONE

Dear Dr. Adhikari Subin,

Thank you for submitting your manuscript to PLOS ONE. After careful consideration, we feel that it has merit but does not fully meet PLOS ONE’s publication criteria as it currently stands. Therefore, we invite you to submit a revised version of the manuscript that addresses the points raised during the review process.

We look forward to receiving your revised manuscript.

Kind regards,

Ahmed A. Al-Karmalawy, PhD

Academic Editor

PLOS ONE

Journal Requirements:

Reviewers' comments:

Reviewer's Responses to Questions

**Comments to the Author**

1. Is the manuscript technically sound, and do the data support the conclusions?

Reviewer #1: Yes

Reviewer #2: Yes

Reviewer #3: Partly

2. Has the statistical analysis been performed appropriately and rigorously? 

Reviewer #1: Yes

Reviewer #2: N/A

Reviewer #3: I Don't Know

3. Have the authors made all data underlying the findings in their manuscript fully available?

Reviewer #1: Yes

Reviewer #2: Yes

Reviewer #3: Yes

4. Is the manuscript presented in an intelligible fashion and written in standard English?

Reviewer #1: Yes

Reviewer #2: Yes

Reviewer #3: Yes

5. Review Comments to the Author

Reviewer #1: Dear Dr Jhashanath Adhikari Subin,

I am grateful for the opportunity to review your manuscript titled "AI-Assisted Discovery of Potent FGFR1 Inhibitors via Virtual Screening and In Silico Analysis" (Manuscript ID: PONE-D-25-21397). Your study is a robust and timely integration of AI-aided virtual screening, molecular docking, and molecular dynamics simulations for the discovery of potential FGFR1 inhibitors. I am thankful for the scope of work and the relevance of the study towards ongoing drug discovery.

Below are some remarks to make your paper tighter, firstly for Abstract (Lines 20–44):

The abstract is well written and informative in all but a few respects. Let me say, however, that I recommend tightening structure for even more clarity and grammatical accuracy in certain places. For example, the phrasing "comparable or slightly lower to that of the native ligand" should be rewritten as "comparable to or slightly lower than that of the native ligand.

secondly for the 'Docking and Interaction Analysis' section (Lines 116–185)

The analysis of interaction can explicitly define hydrogen bonding and hydrophobic contacts. To provide a better indication of the ligand–protein interactions, I would suggest including other relevant non-covalent interactions (e.g., π–π stacking, cation–π, halogen bonding), if your analysis includes any. thirdly for section 'Similarity & pIC₅₀ Predictions' (Lines 399–431) The predicted pIC₅₀ values (7.07–7.47) are promising, though reflecting lead-like instead of drug-like potency. Please consider clarifying that these compounds represent early-stage candidates with room for further optimization.

finally for the section 'Conclusion' (Lines 439–455)

While the conclusion is good and well in accordance with your results, there is repetition of information already described in the abstract. Cutting this section back and emphasizing future directions (e.g., experimental confirmation) would make it more useful.

Overall Assessment

Your manuscript is a valuable contribution to the field of AI-aided drug discovery. The workflow is well done, the methodology is acceptable, and the findings are timely and relevant. Minor revisions to improve clarity and expand discussion in regions where they can improve accuracy further, your manuscript will be ready for publication.

I strongly recommend publication of your manuscript upon minor revisions.

Reviewer #2: This manuscript reported a study that integrates artificial intelligence, virtual screening, and computational modeling to discover novel inhibitors of Fibroblast Growth Factor Receptor 1 (FGFR1). Given the important role of FGFR1 in cancer therapy, the topic is highly relevant. The research is generally well-executed, and the findings are encouraging. However, the manuscript would be suitable for publication after addressing the following minor revisions.

1. Throughout the manuscript, check in typo error and formatting of text.

2. Though the combination of AI and virtual screening is becoming more common, the authors need to more clearly explain what sets their approach apart from existing studies on FGFR1 inhibitors, if such studies have been previously reported?

3. Is there a specific reason why only the 4ZSA PDB structure was selected for this study?

4. The authors should provide a higher-resolution version of Figure 10.

5. What is the synthetic accessibility of the compounds M34, M39, M26, M32, and M28?

Reviewer #3: The efforts exerted in this current work are so appreciated. However, some points need to be addressed. So, a major revision may be required to improve the manuscript:

1. Abbreviations should be defined for the first time (e.g. CID)

2. Regarding the protein selected for molecular docking with PDB: 4ZSA, the structure of the native ligand from website (4UT) is totally different from that afforded in the manuscript.

3. Regarding MD simulations, MM-GBSA is recommended to be calculated.

4. Regarding MD simulations, heat maps and amino acid interaction histograms should be afforded.

5. The conclusion should be more concise and more effective.

6. Some references need to be updated.

7. The resolution of some figures needs improvement.

6. PLOS authors have the option to publish the peer review history of their article (what does this mean? ). If published, this will include your full peer review and any attached files.

**Do you want your identity to be public for this peer review?** For information about this choice, including consent withdrawal, please see our Privacy Policy .

Reviewer #1: **Yes**

Reviewer #2: No

Reviewer #3: No

---

## [Author Response · Author response to Decision Letter 1]

10 Aug 2025

Point-by-Point Response to Reviewer’s Comments

Dear Editor,

We would like to thank you and the reviewers for the constructive and valuable feedback. We have attempted to incorporate all the suggestions to the best of knowledge and skills. It has helped to fix the errors and fulfill the insufficiencies to uplift the scientific content of the manuscript.

Please let us know in case the responses are unclear or incorrect.

The point-by-point responses are mentioned below.

Sincerely,

Authors

Editor Comments:

Journal Requirements:

Response: I have reviewed the PLOS ONE style guidelines and updated the manuscript files to comply with the formatting and file naming requirements. Please let me know if any further adjustments are needed.

Response: No new code was generated for this study. Molecular dynamics simulations were conducted using standard GROMACS protocols as outlined in the GROMACS manual and tutorials by Justin Lemkul (http://www.mdtutorials.com/gmx/complex/index.html). For machine learning, publicly available code and packages from TeachOpenCADD (https://projects.volkamerlab.org/teachopencadd/) were used.

Response: We confirm that the manuscript includes the key findings, along with the md.mdp file for MDS and additional relevant information provided in the Supplementary Information file.

Reviewer’s Comments:

Reviewer #1:

I am grateful for the opportunity to review your manuscript titled "AI-Assisted Discovery of Potent FGFR1 Inhibitors via Virtual Screening and In Silico Analysis" (Manuscript ID: PONE-D-25-21397). Your study is a robust and timely integration of AI-aided virtual screening, molecular docking, and molecular dynamics simulations for the discovery of potential FGFR1 inhibitors. I am thankful for the scope of work and the relevance of the study towards ongoing drug discovery.

Below are some remarks to make your paper tighter:

1. Firstly, for Abstract (Lines 20-44):

The abstract is well written and informative in all but a few respects. Let me say, however, that I recommend tightening structure for even more clarity and grammatical accuracy in certain places. For example, the phrasing "comparable or slightly lower to that of the native ligand" should be rewritten as "comparable to or slightly lower than that of the native ligand".

Response: We agree with your suggestion and have revised the sentence accordingly in the abstract section.

2. Secondly for the 'Docking and Interaction Analysis' section (Lines 116-185)

The analysis of interaction can explicitly define hydrogen bonding and hydrophobic contacts. To provide a better indication of the ligand–protein interactions, I would suggest including other relevant non-covalent interactions (e.g., π-π stacking, cation-π, halogen bonding), if your analysis includes any.

Response: The initial interaction analysis using LigPlot identified only hydrogen bonds and hydrophobic interactions. To address this limitation, we re-analyzed the complexes using PLIP, which enabled the identification of additional interaction types, including π-π stacking and halogen bonds. Accordingly, the interaction analysis section in the manuscript has been revised, replacing the LigPlot-derived results with those obtained from PLIP.

3. Thirdly for section 'Similarity & pIC₅₀ Predictions' (Lines 399-431):

The predicted pIC₅₀ values (7.07-7.47) are promising, though reflecting lead-like instead of drug-like potency. Please consider clarifying that these compounds represent early-stage candidates with room for further optimization.

Response: We have revised the section as suggested to clarify that the predicted pIC₅₀ values indicate lead-like potency and that these compounds represent early-stage candidates with potential for further optimization.

4. Finally for the section 'Conclusion' (Lines 439–455):

While the conclusion is good and well in accordance with your results, there is repetition of information already described in the abstract. Cutting this section back and emphasizing future directions (e.g., experimental confirmation) would make it more useful.

Response: We have shortened the conclusion to avoid repetition with the abstract and have emphasized future directions, including experimental validation and lead optimization.

Reviewer #2:

This manuscript reported a study that integrates artificial intelligence, virtual screening, and computational modeling to discover novel inhibitors of Fibroblast Growth Factor Receptor 1 (FGFR1). Given the important role of FGFR1 in cancer therapy, the topic is highly relevant. The research is generally well-executed, and the findings are encouraging. However, the manuscript would be suitable for publication after addressing the following minor revisions.

1. Throughout the manuscript, check in typo error and formatting of text.

Response: The manuscript has been thoroughly checked for typographical errors and formatting inconsistencies, and necessary corrections have been made.

2. Though the combination of AI and virtual screening is becoming more common, the authors need to more clearly explain what sets their approach apart from existing studies on FGFR1 inhibitors, if such studies have been previously reported?

Response: This study uniquely applies a voting classifier combining three ML models for FGFR1 inhibitor prediction, an approach not previously reported for FGFR1, though used in our earlier work on 3CLpro. Moreover, we screened ~10 million compounds, representing the largest ML-based screening effort for FGFR1 to date, distinguishing our study from existing ones.

3. Is there a specific reason why only the 4ZSA PDB structure was selected for this study?

Response: We selected 4ZSA for its high resolution, native ligand presence, and well-defined active site, making it ideal for accurate docking and comparison.

4. The authors should provide a higher-resolution version of Figure 10.

Response: The original figure was created at 650 dpi and has now been updated to a higher resolution of 1200 dpi as requested which is now assigned as Fig. 15.

5. What is the synthetic accessibility of the compounds M34, M39, M26, M32, and M28?

Response: All the mentioned compounds (M34, M39, M26, M32, and M28) are synthetically accessible. Their availability and sourcing details, including information on chemical vendors, can be found in the PubChem database under the section titled Chemical Vendors for each respective compound.

Reviewer #3:

The efforts exerted in this current work are so appreciated. However, some points need to be addressed. So, a major revision may be required to improve the manuscript:

1. Abbreviations should be defined for the first time (e.g. CID)

Response: We have revised the manuscript to define all abbreviations, including CID (Compound Identifier), at their first occurrence for clarity.

2. Regarding the protein selected for molecular docking with PDB: 4ZSA, the structure of the native ligand from website (4UT) is totally different from that afforded in the manuscript.

Response: For your information, the original 4ZSA structure had missing amino acid residues, so we performed homology modeling using Swiss-Model where 5B7V.1.A template was chosen which contains the native ligand LWJ rather than 4UT. This modeling approach and the rationale for selecting the 5B7V.1.A template are detailed in the Methods section (2.3 Molecular Docking Calculations).

3. Regarding MD simulations, MM-GBSA is recommended to be calculated.

Response: Thank you for the suggestion. We used MMPBSA as it is a more updated method that generally provides more accurate binding free energy estimates than MMGBSA. MMPBSA uses the Poisson-Boltzmann equation, offering a more rigorous treatment of electrostatics and solvation effects in complex biological systems. Therefore, MMGBSA calculations were not performed.

4. Regarding MD simulations, heat maps and amino acid interaction histograms should be afforded.

Response: Decomposition analysis of binding free energy and its analysis was performed. Per-residue free energy change contribution bar plots and heatmaps for the 20 ns equilibrated segment were generated. Additionally, amino acid interaction heatmaps and histograms covering the full 200 ns (20,000 frames) were created as suggested and are provided in the Supplementary Information.

5. The conclusion should be more concise and more effective.

Response: We have revised the conclusion to make it more concise and focused, reducing repetition while highlighting the key findings and clearly outlining future directions.

6. Some references need to be updated.

Response: Since no specific references were indicated, we updated several citations with recent literature where appropriate, while retaining key original studies due to their foundational relevance. Older references related to software were kept as they remain current.

7. The resolution of some figures needs improvement.

Response: We have reviewed all figures and updated their resolution to 600 dpi to improve clarity.

---

## [Decision Letter · Decision Letter 1]

22 Aug 2025

AI-Assisted Discovery of Potent FGFR1 Inhibitors via Virtual Screening and In Silico Analysis

PONE-D-25-21397R1

Dear Dr. Subin,

We’re pleased to inform you that your manuscript has been judged scientifically suitable for publication and will be formally accepted for publication once it meets all outstanding technical requirements.

Kind regards,

Ahmed A. Al-Karmalawy, PhD

Academic Editor

PLOS ONE

Additional Editor Comments (optional):

Reviewers' comments:

Reviewer's Responses to Questions

**Comments to the Author**

1. If the authors have adequately addressed your comments raised in a previous round of review and you feel that this manuscript is now acceptable for publication, you may indicate that here to bypass the “Comments to the Author” section, enter your conflict of interest statement in the “Confidential to Editor” section, and submit your "Accept" recommendation.

Reviewer #1: All comments have been addressed

Reviewer #2: All comments have been addressed

2. Is the manuscript technically sound, and do the data support the conclusions?

Reviewer #1: Yes

Reviewer #2: Yes

3. Has the statistical analysis been performed appropriately and rigorously? 

Reviewer #1: Yes

Reviewer #2: N/A

4. Have the authors made all data underlying the findings in their manuscript fully available?

Reviewer #1: Yes

Reviewer #2: Yes

5. Is the manuscript presented in an intelligible fashion and written in standard English?

Reviewer #1: Yes

Reviewer #2: (No Response)

6. Review Comments to the Author

Reviewer #1: (No Response)

Reviewer #2: All reviewer comments and suggestions have been thoroughly addressed, and the manuscript has been revised accordingly. I believe it now meets all the required standards and is ready for acceptance.

7. PLOS authors have the option to publish the peer review history of their article (what does this mean? ). If published, this will include your full peer review and any attached files.

**Do you want your identity to be public for this peer review?** For information about this choice, including consent withdrawal, please see our Privacy Policy .

Reviewer #1: No

Reviewer #2: No

---

## [Editor Report · Acceptance letter]

PONE-D-25-21397R1

PLOS ONE

Dear Dr. Adhikari Subin,

I'm pleased to inform you that your manuscript has been deemed suitable for publication in PLOS ONE. Congratulations! Your manuscript is now being handed over to our production team.

Kind regards,

on behalf of

Associate Professor Ahmed A. Al-Karmalawy

Academic Editor

PLOS ONE